# A Comparison of Unmodified and Sawdust Derived-Cellulose Nanocrystals (CNC)-Modified Polyamide Membrane Using X-ray Photoelectron Spectroscopy and Zeta Potential Analysis

**DOI:** 10.3390/polym15010057

**Published:** 2022-12-23

**Authors:** Amos Adeniyi, Danae Gonzalez-Ortiz, Celine Pochat-Bohatier, Sandrine Mbakop, Maurice S. Onyango

**Affiliations:** 1Department of Chemical, Metallurgical and Materials Engineering, Tshwane University of Technology, Pretoria 0001, South Africa; 2Water for Rural Communities (WARUC) NPC, Pretoria 0002, South Africa; 3Institut Européen des Membranes, IEM UMR-5635, Université de Montpellier, ENSCM, CNRS, Place Eugène Bataillon, CEDEX 5, 34095 Montpellier, France

**Keywords:** CNC, polyamide, membrane, zeta potential, XPS, interfacial polymerization

## Abstract

Cellulose nanocrystals (CNC) obtained from waste sawdust were used to modify the polyamide membrane fabricated by interfacial polymerization of m-phenylene-diamine (MPDA) and trimesoyl chloride (TMC). The efficiency of the modification with sawdust-derived CNC was investigated using zeta potential and X-ray photoelectron spectroscopy (XPS). The effect of the modification on membrane mechanical strength and stability in acidic and alkaline solutions was also investigated. Results revealed that the negative zeta potential decreased at a high pH and the isoelectric point shifted into the acidic range for both modified and unmodified membranes. However, the negative charges obtained on the surface of the modified membrane at a pH lower than 8 were higher than the pristine membrane, which is an indication of the successful membrane modification. The XPS result shows that the degree of crosslinking was lowered due to the presence of CNC. Enhanced stability in solution in all pH ranges and the increase in mechanical strength, as indicated by higher Young’s modulus, maximum load, and tensile strength, confirmed the robustness of the modified membrane.

## 1. Introduction

The success of membrane technology in water treatment has been reported worldwide [1]. However, some drawbacks in the synthesis of membranes affecting their performances required further investigation. In some cases, a membrane with a high pure water flux might have a very low rejection rate. Depending on the type of membrane being used, specific solute rejection is affected by either repulsion or attraction between the membrane surface and a specific solute in water [2]. For instance, the best membranes for high-pressure water treatment procedures are thin-film composite (TFC) membranes [3]. It usually has a high rejection rate for both organic and inorganic solutes in all varieties of water matrix [4]. The limitations of this membrane include a trade-off between water flux and solute rejection, fouling, susceptibility to chlorine attack, and low shelf life [5,6,7]. Membrane treatment is needed in harsh environments that include mine wastewater treatment. Therefore, thin-film composite membranes require further improvement to increase water recovery while maintaining their physiochemical properties, which include high solute rejection ability, mechanical strength, and stability even in harsh environments [8]. One effective method for enhancing the performance and longevity of the membrane has been demonstrated to be the insertion of suitable nanoparticles on its surface [9,10,11,12]. Nanomaterials with rational functionality, superparamagnetic characteristics, antibacterial capabilities, high hydrophilicity, and strong hydration capacities may be better suited for membrane modification [13]. As a result, several nanomaterials have been reportedly used to modify TFC membranes [14,15].

In our previous study, the use of cellulose nanocrystals derived from sawdust to modify polyamide membranes was reported [16]. This resulted in an increase in water flux and hydrophilicity while a high rejection of solute is maintained. The interest in sawdust-derived cellulose nanocrystals is provoked not only by the properties of the nanomaterial but also by its abundant availability and environmental compatibility [17]. Improved mechanical strength and hydrophilicity are very important for an efficient membrane that can withstand a harsh environment [18]. It has been noted that cellulose nanocrystals serve as the last line of defense that prevents cracking and upholds the integrity of the membrane [19]. For instance, the membrane’s response to stress in terms of deformation determines the mechanical power of the membrane [20]. A membrane may typically undergo elastic deformation before the resistance limit is reached and exceeded, causing inelastic or irreversible distortion as the force is increased. Avoiding fatigue damage is essential for a membrane used in industrial applications. This is because inelastic deformation due to excessive stress contributes to membrane fatigue damage. CNC integration can improve the membrane’s mechanical strength and prevent failure during operation. The mechanical strength of membranes can be improved through CNC based on their mechanical properties and their capacity to disperse in the polymer solution [21,22]. If CNC is evenly distributed across polymer solutions, surface interactions such as the van der Waals and hydration forces can be defeated. As nanoparticles form during the crystallization of the polymer, it is possible to achieve strength and load transmission from CNC to composite membranes.

Zeta potential measurement is a procedure that can be used to measure the charge on the surface of membranes and can also be used to predict ionic solute rejection in an aqueous solution [23]. Based on changes in the surface chemistry, the approach may also effectively assess the impact and efficacy of surface modification. This is because the surface chemistry is changed due to modification even though the membrane is not yet in contact with an aqueous solution. XPS is a method used to establish the elemental composition of the material. The method has been used recently to determine the degree of crosslinking in polymeric membranes [24,25]. The degree of crosslinking determines the membrane tightness and, hence, salt rejection and water permeability. Both XPS and zeta potential measurements can be combined to determine the effectiveness of a membrane modification.

Consequently, the aim of this study is to investigate the effectiveness of membrane modification with cellulose nanocrystals using zeta potential analysis and XPS. The work sought to establish improved mechanical strength and stability as a result of the modification of the polyamide membrane. XPS was used to evaluate the degree of crosslinking. Zeta potential analysis and the XPS showed that the modification was effective. The modified membrane gives off a higher negative charge and exhibits a lower degree of crosslinking and higher water permeability. The modified membrane also exhibits stability in highly alkaline and acidic solutions and higher mechanical strength, as indicated by increased Young’s modulus, maximum load, and tensile strength.

## 2. Materials and Methods

The membrane fabrication technique, the modification, and the materials used have been reported in the work published by Adeniyi et al. [16]. The technique used was interfacial polymerization. The monomers were TMC and MPDA; both were obtained from Sigma-Aldrich, Centurion, South Africa. The CNC used for the modification was derived from sawdust. The CNC was obtained from CSIR, Durban, South Africa. After many trials, the membrane with 0.21% CNC was selected for further investigation and comparison with the unmodified membrane. The modified membrane is tagged PA-CNC while the pristine membrane is tagged PA.

### 2.1. Zeta Potential (ZP) Measurement

The Zeta potential measurement was carried out using a Surpass electrokinetic analyzer, made by Anton Paar, Graz, Austria. The equipment is equipped with an adjustable “gap” measuring cell with a measuring surface of 20 mm by 10 mm. The procedure has been described in our previous work [26].

### 2.2. X-ray Photoelectron Spectroscopy (XPS)

X-rays are used in XPS to extract photoelectrons from a surface. Using the X-ray photon energy and the extracted electrons’ kinetic energy, one may calculate the binding energy of the extracted electrons. It is possible to identify the elements from which the electrons were extracted using this one-of-a-kind quantity. The approach can identify all elements (apart from hydrogen and helium), as well as compounds because the binding energy of an element changes from compound to compound. It is primarily a surface technique, as the escape depth of the photoelectrons ranges from 2 to 10 nm. The XPS detection threshold is roughly 0.1%. Sputtering the surface with powerful argon ions while keeping an eye on the binding energy areas of particular elements can provide information from layers underneath the outer surface. The chemical contents of the polyamide layers of the membrane were examined using an X-ray photoelectron spectrometer fitted with Monochromatic Al k (1486.7 eV). The ESCAlab 250Xi model (Thermo, Waltham, MA, USA) was the one used. The experimental setup is shown in Table 1. The XPS spectra were calibrated using an Ag standard to ensure that the binding energy scale was correct.

### 2.3. Mechanical Test

The mechanical test was performed using an Instron tensile tester–5966 manufactured by Instron Engineering Corporation, Norwood, MA, USA. The sample sizes were 10 mm wide and 0.2 mm thick. The membranes were strained at a rate of 1 mm/min. The gauge length was 25 mm.

### 2.4. Stability in the Acidic and Basic Environment

The membranes were submerged in 0.1 L of either 1 M HCl or 1 M NaOH aqueous solution for two weeks at room temperature. After that, the membranes were washed with distilled water until the pH was neutral [27]. The weight change after one week was noted and used as an additional pH-stability indication. The membrane’s performance was subsequently evaluated again for water flux and sodium chloride (1500 ppm) rejection.

### 2.5. Analysis

The equation relating the elemental composition (%) of nitrogen (N) and oxygen (O) to the degree of crosslinking is given by Equation (1) [24] where n is the degree of crosslinking. The ratio N/O is obtained from XPS results.
(1)$$\frac{N}{O}=\frac{n+2}{4-n}\;$$

If the ratio (N/O) is assigned to r, then Equation (1) can be re-written as follows:(2)$$n=\frac{4r-2}{r+1}\;$$

Water flux calculation was performed based on Equation (3) [16]
(3)$$J_{w}=\frac{V}{At}\;$$
where the water flux is Jw (L/m^2^/h) and the permeate volume is V (L). The filtration period t is measured in h, whereas the active membrane area A is measured in m2.

The rejection of solute was calculated based on Equation (4).
(4)$$R=\frac{\left(C_{f}-C_{p}\right)}{C_{f}}\times 100\%\;$$
where C_f_ is the solute concentration in the feed, C_p_ is the solute concentration in the permeate, and R is the rejection percentage.

## 3. Results and Discussion

### 3.1. Surface Charge

Figure 1 shows the ZP pattern for both PA-CNC and PA membranes. Both membranes have surfaces with a predominance of negative charges. The carboxylic group that forms in the membrane because of the interfacial polymerization of highly reactive TMC, an acid chloride monomer with high functionality, with MPDA is what gives the PA membrane its negative charge [28,29]. The addition of CNC, a negatively charged particle, must have increased the negative charge observed in the PA-CNC membrane. The addition of nanomaterials has always been observed to result in a greater negative surface charge of the PA membrane [30]. For both membranes, the negative zeta potential decreased at a high pH. The isoelectric point, the pH where there is no effective charge on the membrane, shifted towards the acidic range. This demonstrates how hydrophilic both membranes are. At a pH range from 4 to 7, the change in apparent zeta potential rose in proportion to the membrane hydrophilicity [31]. At pH levels below 8, the new membrane’s zeta potential was, nevertheless, more negative than the pristine membrane. This suggests that the membrane has been successfully modified. This is also demonstrated by the increased water permeability compared to the original membrane [16]. The higher negative zeta potential of the PA membrane between pH 8 and 10 is because of the dissociation of a more carboxylic group in the membrane [32].

### 3.2. Elemental Composition and the Degree of Crosslinking

The XPS analysis shows the elemental compositions of both the PA and PA-CNC membranes. The outcomes are displayed in Figure 2. The results analysis is displayed in Table 2. The ratio of Nitrogen to Carbon remains unchanged whereas the ratio of oxygen to carbon and nitrogen to oxygen changed. Theoretically, the oxygen atoms for the PA membrane come from carboxylic acid groups cleaved from unreacted acyl chloride groups during interfacial polymerization and the functionality of the –NHCO–bond [28]. The –NHCO bond and the –NH2 end groups are where the nitrogen atoms are found. Equation (2) was used to calculate the degree of crosslinking between the nitrogen and oxygen values acquired from the XPS readings. A higher N/O ratio is an indication of a higher degree of crosslinking. The relative amounts of elements were affected by the modification with CNC (Table 2). The modification increased the relative amount of oxygen and reduced the amount of nitrogen and carbon. For the PA-CNC membrane, the N/O ratio was reduced because of the successful modification of the polyamide with the cellulose nanocrystals. Increasing the concentration of oxygen was attributed to the presence of hydroxyl groups on the CNC, and decreasing the amount of nitrogen after modification is understood as the attenuation of nitrogen from the membrane surface [33]. This means that the structure of the polyamide membrane was changed by the incorporation of CNC. This is because cellulose nanocrystals carry the OH group into the polyamide and thus increase the composition of the atomic oxygen in the modified membrane. The addition of CNC led to a loose polyamide membrane because the N/O ratio of the PA-CNC membrane was significantly lower than that of the PA membrane. Fewer acyl chloride groups on TMC, some of which may have reacted with the CNC, may be the source of the decreased degree of cross-linking. However, the degree of crosslinking obtained with CNC is better than the value reported in the literature [24].

The Na peaks on both membranes located at 1100 eV are observed because sodium hydroxide was added to the aqueous phase as an acid acceptor. Both survey spectra showed the Na KLL Auger peak at 500 eV [34]. The presence of polyamide was confirmed by C 1s, N 1s, and O 1s peaks at 284.1, 398.9, and 530.8 eV for PA membrane and at peaks of 284.2, 399, and 530.9 eV, respectively, for PA-CNC membrane [35]. In the XPS survey profiles, the cellulose signals for C 1s and O 1s are typically 286 eV and 533 eV, respectively [36]. The percentage of O 1s increased from 15.1 to 17.4 after the addition of cellulose nanocrystals, as expected. This is a result of the OH group in the cellulose nanocrystals. The presence of Cl 2p in the modified membrane indicated that the ammonium chloride added for pH adjustment was adequately absorbed in the crosslinking due to the modification with cellulose nanocrystals.

High-resolution C 1s, O 1s, and N 1s scans were performed to obtain more detail on the modification of the polyamide film. The results are shown in Figure 3. For the PA membrane, the C 1s deconvoluted into four sub-peaks attributed to C=O at 287.5 eV, C-O at 285.8 eV, C-C Sp^3^ at 284.8 eV, and C-C Sp^2^ at 284 eV. Similarly, for the PA-CNC membrane, the C 1s were deconvoluted into four sub-peaks attributed to C=O at 287.5 eV, C-O at 286.6 eV, C-C Sp^3^ at 285 eV, and C-C Sp^2^ at 284 eV. The change in the peaks observed for C-O is due to the incorporation of CNC. This is a chemical shift, and it is due to a change in the chemical bonding of C-O as a result of the incorporation of CNC [37]. Core binding energies are determined by the electrostatic interaction between it and the nucleus. They are reduced by the electrostatic shielding of the nuclear charge from all other electrons in the atom (including valence electrons). Moreover, the removal or addition of electronic charge as a result of changes in bonding will alter the shielding. 

The O 1s core spectrum for PA and PA-CNC membranes is shown in Figure 3C,D. The peaks are at 530.6, 532.4, and 535.2 eV for the PA membrane. For the PA-CNC membrane, the peaks are at 530.6, 532.6, and 535.1 eV. All these peaks are assigned to organic O. These peaks mean that the carboxylic groups are present in both membranes. The deconvoluted N 1s core spectrum is shown in Figure 3E,F. The peaks are at 397.8, 399.2, and 402.1 eV for the PA membrane. The peaks are at 397.7, 399.2, and 402.1 eV for the PA-CNC membranes. These peaks are an indication of the presence of an amide group in both the PA and the PA-CNC membranes.

### 3.3. Mechanical Strength of the Membranes

Polymeric membranes are required to have good mechanical durability for their reliability and life span. The mechanical durability can be measured by analyzing membrane real stress loading conditions. The required load is determined by measuring the membrane’s strain at a controlled rate. Typically, metrics such as Young’s modulus, toughness, tensile strength, and elongation are used to measure this. Elongation and toughness are required in addition to Young’s modulus and tensile strength in order to fully describe the mechanical behavior of the membrane [38]. The linear component of the stress-strain isotherm’s first slope, or roughly the first 2.0% of strain, is utilized as an indicator of the stiffness of the elastic deformation initiation known as Young’s module [21]. Tensile strength, which can be used to gauge a membrane’s strength, is the highest stress that a membrane can withstand before breaking during stretching. The ductility of a membrane is determined by its elongation-at-break value. Toughness is defined as how much energy a material can withstand before failing [39]. Figure 4 shows the stress-strain curves. Five specimens were tested for each of the membranes. The curve shows the expected pattern for membranes under the application of stress. The linear part is the region of reversible deformation where the membrane can easily return to its normal form. As the applied stress increases, the membranes enter the region of irreversible deformation, as indicated by the curves. The membranes ultimately reach their fracture strength, after which breakage occurs.

Table 3 shows Young’s modulus, tensile strength, and elongation. The results show that these variables increased because of the presence of cellulose nanocrystals. This may be due to the properties of the cellulose nanocrystals and the fact that they are well dispersed in the aqueous phase monomer solution. CNC had a large ratio of length to diameter and excellent mechanical characteristics [40]. An interactive network was developed by the interactions between the CNC and the MPDA monomer due to hydrophilic hydroxyl groups in the CNC [29]. This interaction creates a bonding (hydrogen) and an electrostatic attraction, that overcomes the van der Waals and the hydration forces, resulting in improved overall mechanical properties of composite membranes [41].

Young’s modulus, a measure of the membrane’s stiffness, was found to increase by 12% due to the presence of CNC. A similar result was obtained by Bai et al. (2017). The tensile strength increases by 13.8%. Mao et al. [40] obtained similar results when 3% of CNC was added to the chitosan membrane. They recorded a 13.2% increase in tensile strength as a result of the addition of CNC. Jahan et al. [42] obtained a similar increase in tensile strength when CNC was added to poly (vinyl alcohol) (PVA) nanocomposite membranes. The elongation was found to reduce from 27.6% for the pristine membrane to 26.8% for the modified membrane.

### 3.4. Stability in the Acidic and Alkaline Environment

No significant swelling was recorded for both membranes after putting them in the acidic solutions for two weeks, as shown in Table 4. For the PA membrane, the weight gain was 0.0035 g, while the weight gain for the PA-CNC was 0.0010. A significant swelling was observed for both membranes when immersed in a 1 M NaOH solution for 2 weeks. The weight gain for PA was 0.0279 g, while that of PA-CNC was 0.0182 g. The water flux at 7 bar increases for both membranes, with the highest increase experienced with the PA membrane, as shown in Figure 5. Moreover, both membranes experience a reduction of sodium chloride rejection, particularly in an alkaline environment, as shown in Figure 6. This is basically due to the hydrolysis of the carboxylic group because of exposure to nucleophilic attack under an alkaline and acidic environment [43]. Most polyamide membranes are stable between a pH of 2 and 11. The stability test was conducted at a pH of acidic and basic ranges. The effect of the alkaline and acidic environment is more significant with the PA membrane. In all cases, the presence of CNC enhanced the membranes’ stability. This may be because the CNC reacted with the carboxylic group and hindered further hydrolysis.

## 4. Conclusion

In this study, the successful interfacial polymerization-fabricated polyamide membrane modified with CNC obtained from sawdust was tested. Zeta potential analysis and X-ray photoelectron spectroscopy (XPS) were used for the verification of the modification. The XPS demonstrated the CNCs presence that caused the degree of crosslinking to decrease. The CNC membrane had a stronger negative charge. The addition of the CNC made the polyamide mechanically stronger by raising its maximum load, tensile strength, and Young’s modulus. Due to the moderate increases in the water flux and decreases in salt chloride rejection, the stability of the membrane was improved while using a polyamide membrane with CNC. The modification of the polymeric membrane with CNC produced from sawdust has significant promise for industrial applications.

## Figures and Tables

**Figure 1 polymers-15-00057-f001:**
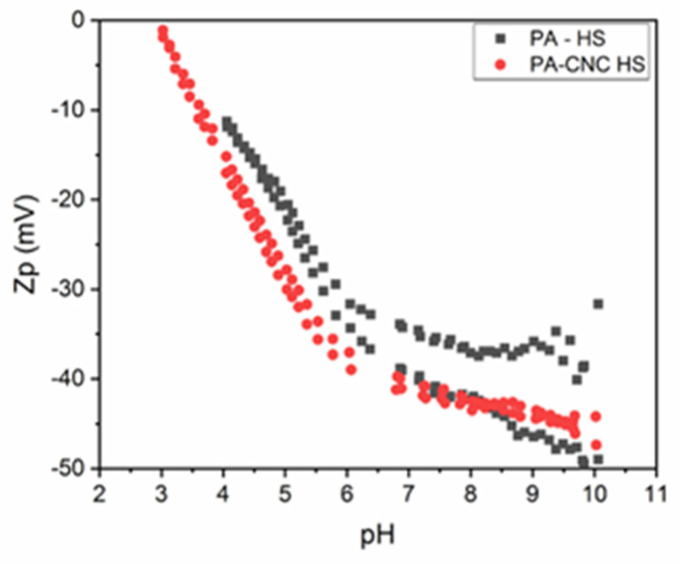
Zeta potential for both the modified (PA-CNC) and the unmodified (PA) membranes.

**Figure 2 polymers-15-00057-f002:**
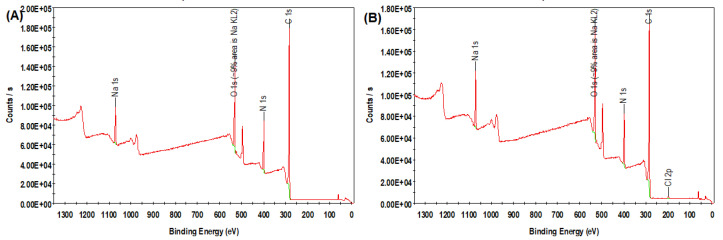
The XPS spectra for both the unmodified membrane (**A**) and the modified membrane (**B**).

**Figure 3 polymers-15-00057-f003:**
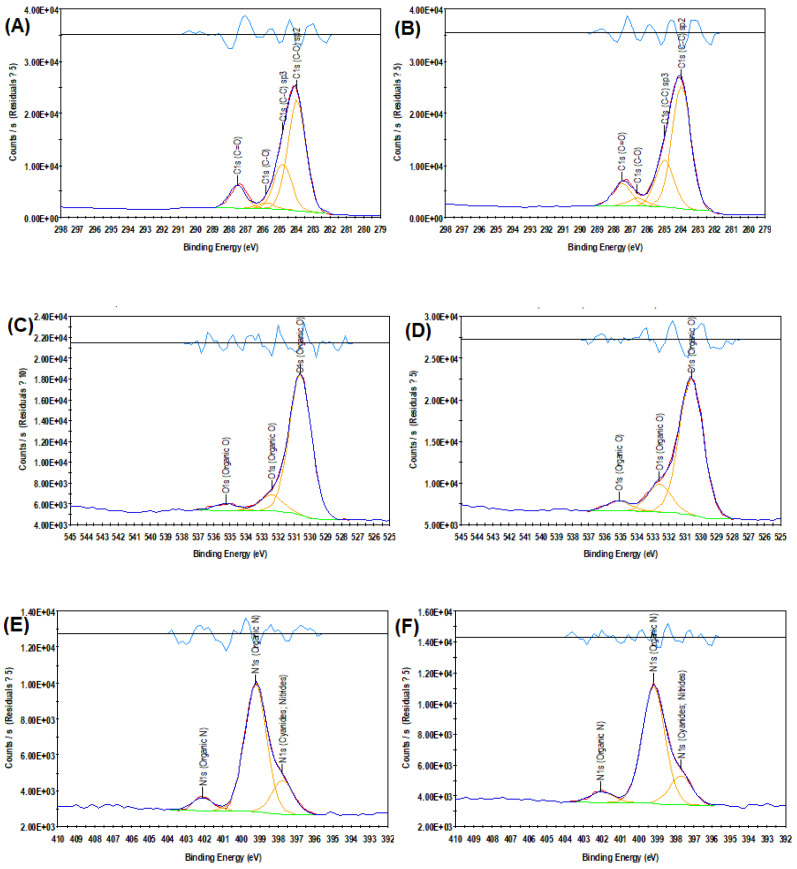
C 1s spectra of the unmodified membrane (**A**), C 1s spectra of the modified membrane (**B**), O 1s spectra of the unmodified membrane (**C**), O 1s spectra of the modified membrane (**D**), N 1s spectra of the unmodified membrane (**E**), N 1s spectra of the modified membrane (**F**).

**Figure 4 polymers-15-00057-f004:**
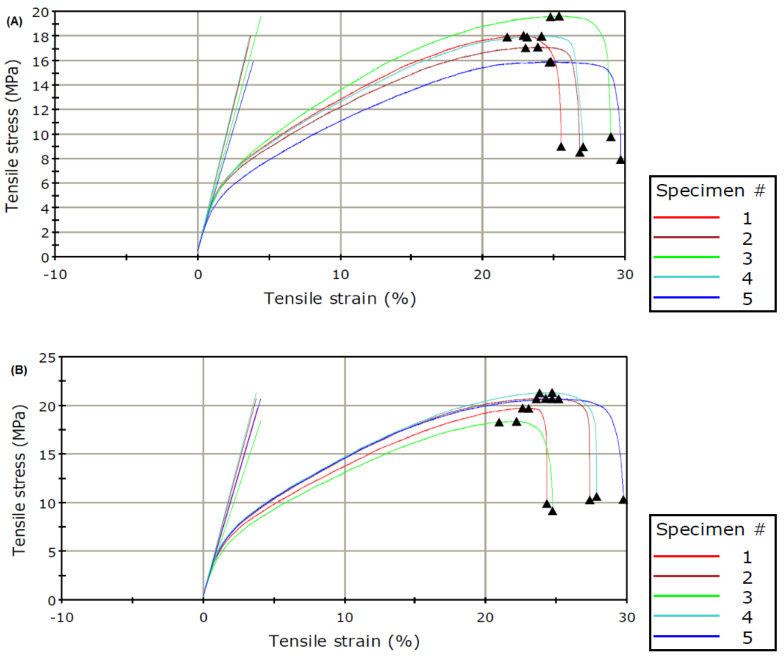
Stress-strain curves for the PA (**A**) and PA-CNC (**B**) membranes. The Figures shows a higher average tensile stress for PA-CNC than the value obtained for PA.

**Figure 5 polymers-15-00057-f005:**
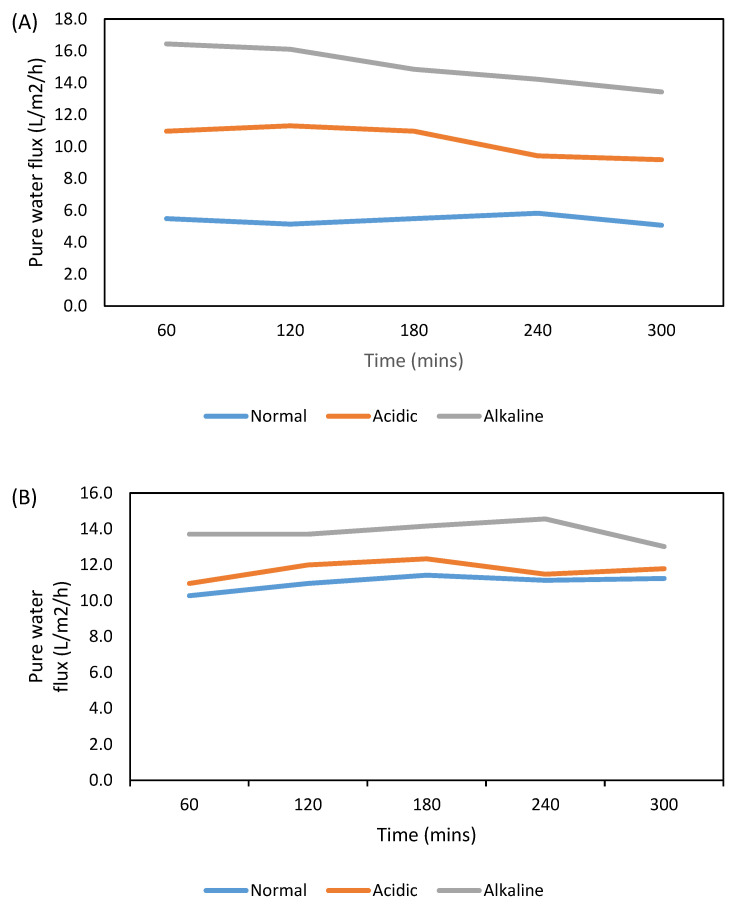
Water flux for PA (**A**) and PA-CNC (**B**) at 7 bar after soaking for two weeks in 1 M HCl (acidic) and 1 M NaOH (alkaline) solutions.

**Figure 6 polymers-15-00057-f006:**
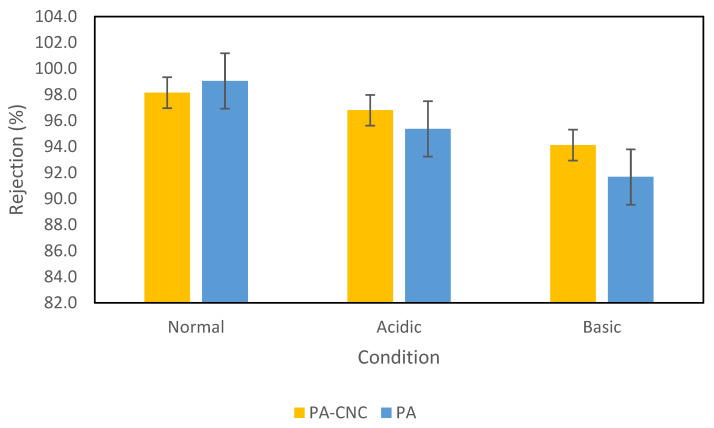
Sodium chloride (1500 ppm) rejection for the membranes where normal is for the membrane before soaking, Acidic is for after soaking in 1 M HCl and alkaline is for after soaking in 1 M NaOH.

**Table 1 polymers-15-00057-t001:** Experimental setup for X-ray photoelectron spectroscopy.

Parameter	Value
Instrument Brand	Thermo, Waltham, MA, USA
Model	ESCAlab 250Xi
X-rays	Monochromatic Al k (1486.7 eV)
X-ray Power	300 W
X-ray Spot Size	900
Pass Energy (Survey)	100 eV
Pass Energy (Hi-res)	20 eV
Pressure	<10^−8^ mBar

**Table 2 polymers-15-00057-t002:** XPS analysis for both the PA and PA-CNC membranes.

Membrane	Elemental Composition			O/C	N/C	r	n
	C	O	N				
PA	69.3	15.1	12.7	0.23	0.18	0.84	0.74
PA-CNC	67.1	17.4	11.8	0.26	0.18	0.68	0.42

**Table 3 polymers-15-00057-t003:** Mechanical characterization of both PA and PA-CNC membranes.

Membrane	Modulus	Maximum	UTS	Elongation
	(MPa)	Load (N)	(MPa)	(%)
PA	450.6 ± 35.7	35.5 ± 2.7	17.7 ± 1.4	27.6 ± 1.7
PA-CNC	505.8 ± 45.5	40.4 ± 2.3	20.2 ± 1.2	26.8 ± 2.3

**Table 4 polymers-15-00057-t004:** Change in weight of both PA and PA-CNC membranes after immersion in 1M NaOH and 1M HCl for two weeks.

**Membrane**	**Initial Weight (g)**	**Final Weight after Immersion in Acid Solution (g)**
PA	0.1940	0.1975
PA-CNC	0.1923	0.1933
**Membrane**	**Initial Weight (g)**	**Final Weight after Immersion in Basic Solution.**
PA	0.1972	0.2251
PA-CNC	0.1925	0.2107

## Data Availability

Not applicable.

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
