# Peer review of "A Comparison of Unmodified and Sawdust Derived-Cellulose Nanocrystals (CNC)-Modified Polyamide Membrane Using X-ray Photoelectron Spectroscopy and Zeta Potential Analysis"

_polymers, 2022, doi:10.3390/polym15010057_

Round 1

Reviewer 1 Report

The work reports the modification of PA membrane with CNC to improve its stability and performance. I recommend publication after major revisions. My comments are:

"The addition of CNC led to a loose polyamide membrane because the N/O ratio of the PA-CNC membrane was significantly lower than that of the PA membrane" well, I don't think this is an accurate statement. XPS is surface technique so the reduction in N/O ration could be attributed to the presence of CNT atop of PA layer. Could you show SEM images to gain insight about whether the CNT are located atop of PA layer or embedded inside it. Also, XPS depth profile may provide useful information here. 

High resolution spectra of membranes in carbon region showed shift in position of peaks. For example, C-O for PA membrane is located 285.8 eV while that for PA-CNC membrane is located at 286.6 eV. what is the reason for this ship. Did the authors calibrate the XPS spectra? Please provide information on this.  This sentence "The change in the peaks observed 191 for C-O is due to the incorporation of CNC" is inaccurate, and further explanation is needed. Also, XPS residual standard deviation   seems high in Figures 3. 

Author mentioned in the introduction: "The limitations of this membrane include a trade-off between water flux and solute rejection" this trade-off relation has been addressed by Al Mayyahi et. al (2018) (Journal of Environmental Chemical Engineering 6 (2018) 1109–1117 ; https://doi.org/10.1016/j.jece.2018.01.035). This work should be mentioned clearly and discussed. 

 More references should be cited. I suggest citing the following papers:

Membranes 20188(3), 66; https://doi.org/10.3390/membranes8030066

Membranes 20188(3), 68; https://doi.org/10.3390/membranes8030068

Environ Chem Lett 16, 1469–1475 (2018). https://doi.org/10.1007/s10311-018-0758-z

Author Response

"The addition of CNC led to a loose polyamide membrane because the N/O ratio of the PA-CNC membrane was significantly lower than that of the PA membrane" well, I don't think this is an accurate statement. XPS is surface technique so the reduction in N/O ration could be attributed to the presence of CNT atop of PA layer. Could you show SEM images to gain insight about whether the CNT are located atop of PA layer or embedded inside it. Also, XPS depth profile may provide useful information here.  Yes I agree that the reduction of N/O could be due to the presence of CNC at the surface of the membrane. However this is not the argument here. The reduction was observed. The implication is that it led to reduction of the degree of crosslinking causing a relatively loose membrane.  

High resolution spectra of membranes in carbon region showed shift in position of peaks. For example, C-O for PA membrane is located 285.8 eV while that for PA-CNC membrane is located at 286.6 eV. what is the reason for this ship. This is a chemical shift, that is, a change in binding energy of a core electron of an element due to a change in the chemical bonding of that element. This is a chemical shift, and it is due to a change in the chemical bonding of C-O as a result of the modification.

Did the authors calibrate the XPS spectra? Please provide information on this.  The XPS spectra is calibrated using a Ag standard to ensure that the Binding energy scale is correct.

This sentence "The change in the peaks observed 191 for C-O is due to the incorporation of CNC" is inaccurate, and further explanation is needed.  Explanation given

Also, XPS residual standard deviation   seems high in Figures 3.  Not necessarily

Author mentioned in the introduction: "The limitations of this membrane include a trade-off between water flux and solute rejection" this trade-off relation has been addressed by Al Mayyahi et. al (2018) (Journal of Environmental Chemical Engineering 6 (2018) 1109–1117 ; https://doi.org/10.1016/j.jece.2018.01.035). This work should be mentioned clearly and discussed. 

 More references should be cited. I suggest citing the following papers:

Membranes 20188(3), 66; https://doi.org/10.3390/membranes8030066

Membranes 20188(3), 68; https://doi.org/10.3390/membranes8030068

Environ Chem Lett 16, 1469–1475 (2018). https://doi.org/10.1007/s10311-018-0758-z

Done

Reviewer 2 Report

1- The way that the authors use abbreviations is confusing. Normally one writes out the name and adds the abbreviation in brackets when the abbreviation is used the first time. From the second time onwards only, the abbreviation is used. The authors of this paper use abbreviations without defining them, and only later in the text the abbreviation is defined. For example, Lines 11, 78, etc.

2- Please consider the significant digits of values, e.g., in Table 4, 0.1940 and 0.1975 should be 0.19 and 0.2, respectively, or 0.1923 and 0.1923 should be 0.19 and 0.19 respectively. The same things are true also in Table 4.

3- Error bars are required for Figure 6 to allow assessment of any perceived differences.

4- Fig 2, what is the peak that appeared at 500 eV related to?

5- The discussion and results are weak.  For example, the XPS section should be strengthened by citing the relevant articles such as:

https://doi.org/10.1016/j.ijbiomac.2017.08.136

https://doi.org/10.1016/j.carbpol.2022.119910

https://doi.org/10.1016/j.carbpol.2021.118550

Author Response

1- The way that the authors use abbreviations is confusing. Normally one writes out the name and adds the abbreviation in brackets when the abbreviation is used the first time. From the second time onwards only, the abbreviation is used. The authors of this paper use abbreviations without defining them, and only later in the text the abbreviation is defined. For example, Lines 11, 78, etc. Corrected

2- Please consider the significant digits of values, e.g., in Table 4, 0.1940 and 0.1975 should be 0.19 and 0.2, respectively, or 0.1923 and 0.1923 should be 0.19 and 0.19 respectively. The same things are true also in Table 4. This is deliberate because of small difference in the values that may be swept off by rounding off the values.

3- Error bars are required for Figure 6 to allow assessment of any perceived differences. Done

4- Fig 2, what is the peak that appeared at 500 eV related to? 500 eV is likely related to the Na KLL Auger peak.  

5- The discussion and results are weak.  For example, the XPS section should be strengthened by citing the relevant articles such as:

https://doi.org/10.1016/j.ijbiomac.2017.08.136

https://doi.org/10.1016/j.carbpol.2022.119910

https://doi.org/10.1016/j.carbpol.2021.118550

Done

Round 2

Reviewer 1 Report

The authors have adequately addressed my comments and suggestions; therefore, I recommend publication.

Author Response

Nothing to respond to.

Reviewer 2 Report

-        Error bars are required for Figure 6 to allow assessment of any perceived differences.

-        On lines 171 and 172, you must use small letters for Oxygen, Nitrogen, and Carbon.

-        On lines 82 and 83, X-ray photoelectron spectroscopy (XPS) should be abbreviated. The way that the authors use abbreviations is confusing. Normally one writes out the name and adds the abbreviation in brackets when the abbreviation is used the first time. From the second time onwards only, the abbreviation is used. The authors of this paper use abbreviations without defining them, and only later in the text the abbreviation is defined.

Author Response

-        Error bars are required for Figure 6 to allow assessment of any perceived differences. Done

Without error bars

With error bars

-        On lines 171 and 172, you must use small letters for Oxygen, Nitrogen, and Carbon. Done

-        On lines 82 and 83, X-ray photoelectron spectroscopy (XPS) should be abbreviated. The way that the authors use abbreviations is confusing. Normally one writes out the name and adds the abbreviation in brackets when the abbreviation is used the first time. From the second time onwards only, the abbreviation is used. The authors of this paper use abbreviations without defining them, and only later in the text the abbreviation is defined. Done
